# Adenoviral Vectors: Potential as Anti-HBV Vaccines and Therapeutics

**DOI:** 10.3390/genes13111941

**Published:** 2022-10-25

**Authors:** Tasneem Farhad, Keila Neves, Patrick Arbuthnot, Mohube Betty Maepa

**Affiliations:** Wits/SAMRC Antiviral Gene Therapy Research Unit, Faculty of Health Sciences, University of the Witwatersrand, Johannesburg 2000, South Africa

**Keywords:** adenovirus, gene therapy, hepatitis B virus, vaccine, viral vectors

## Abstract

Adenoviral vaccines have been at the front line in the fight against pandemics caused by viral infections such as Ebola and the coronavirus disease 2019. This has revived an interest in developing these vectors as vaccines and therapies against other viruses of health importance such as hepatitis B virus (HBV). Current hepatitis B therapies are not curative; hence, chronic hepatitis B remains the major risk factor for development of liver disease and death in HBV-infected individuals. The ability to induce a robust immune response and high liver transduction efficiency makes adenoviral vectors attractive tools for anti-HBV vaccine and therapy development, respectively. This review describes recent developments in designing adenoviral-vector-based therapeutics and vaccines against HBV infection.

## 1. Introduction

Hepatitis B virus (HBV) infection continues to pose a serious global health problem. It is estimated that 296 million people globally are infected with HBV, with the highest prevalence occurring in regions such as sub-Saharan Africa, and east and southeast Asia [1]. In these endemic and hyperendemic areas, infants and children are the groups most at risk. Perinatal transmission from infected mothers to their newborn babies or horizontal transmission from infected family members to children are two primary mechanisms of HBV infection in these regions. While many infections acquired in adulthood are acute and successfully cleared by the immune system, the vast majority of infections that occur during infancy or early childhood become chronic, which increases the risk of developing life-threatening HBV-associated illnesses, such as cirrhosis and hepatocellular carcinoma (HCC) [2,3,4]. 

HBV belongs to the *Hepadnaviridae* family of viruses and carries a circular, partially double-stranded genome of ~3.2 kb. The genome includes four overlapping open reading frames (ORFs), namely *polymerase* (*P*), *precore/core* (*C*), *surface* (made up of *pre-S1*, *pre-S2*, and *S* regions), and the *X* region [5]. Following binding to the host sodium-taurocholate co-transporting polypeptide (NTCP) receptor and entry into the hepatocyte, viral polymerase converts the partially double-stranded relaxed circular DNA (rc-DNA) into covalently closed circular DNA (cccDNA), which remains in the nucleus as a stable minichromosome [6,7]. Additionally, HBV DNA may integrate into the host genome, which is a contributor to persistent HBV surface antigen (HBsAg) secretion and progression to HCC [8,9,10]. The diagnosis of acute and chronic HBV infection is dependent on the detection of viral biomarkers, with HBsAg being the primary clinical marker for HBV infection [11]. 

The current vaccine against HBV is made up of a recombinant HBV small surface antigen (S-HBsAg) and offers safe and effective protection against HBV infection [12]. However, this protein vaccine is less effective in those older than 40 years and in immune-compromised individuals [13,14]. As a result of the Coronavirus disease 2019 (COVID-19) pandemic, old and new vaccine strategies such as protein, inactivated viral strains, mRNA, and adenoviral vector (AdV)-based technologies have been extensively explored and offer promising outcomes [15,16]. Protein and inactivated strain vaccines have generally resulted in a lower overall protective efficacy in recent human trials as compared to nucleic-acid-based vaccines [17,18,19,20]. Nucleic-acid-based vaccines induce a more durable and broader immune response, with mRNA vaccines generally inducing the highest overall protective efficacy [17,19,20]. A strong innate response induced by AdVs makes them particularly interesting for vaccine development. 

Available antiviral treatments include nucleotide/nucleoside analogs (NAs), which prevent HBV DNA synthesis, and immunomodulatory interferons (IFNs) [21]. Although these antivirals play a role in keeping viral infection under control, they rarely achieve complete virological cure because they do not act on the stable cccDNA. Use of gene therapy to impair viral replication using HBV-specific gene silencing, gene editing, and epigenome modifications has been a promising strategy. Common approaches to HBV gene silencing/editing involve the use of RNA interference (RNAi), transcription-activator-like effector nucleases (TALENs), or clustered regularly interspaced short palindromic repeats (CRISPRs) and CRISPR-associated protein (CRISPR/Cas) systems [22]. Despite the availability of potent anti-HBV gene therapeutics, finding safe and efficient delivery methods is the major challenge to clinical translation. Viral vectors such as lentiviral vectors, adeno-associated viral vectors (AAVs), and AdVs have shown promise in gene therapy [23]. AdVs are valuable for targeting HBV because of their inherent hepatotropism. 

## 2. Adenoviruses as Vectors

Adenoviruses (Ads) belong to a family of *Adenoviridae* and are subdivided into five genera, of which human Ads fall under the *mastadenoviridae*. They are further subdivided into seven species named A to G, with almost 70 serotypes that can infect humans. Widely studied species C serotype 2 (Ad2) and serotype 5 (Ad5) cause infections of the upper respiratory system. Ads are non-enveloped, double-stranded DNA viruses with a genome of around 26 to 45 kb. The genome comprises two groups of genes that are either expressed before (early) or after (late) viral DNA replication. Among other functions, the early units (E1 to E4) encode proteins essential for viral genome replication, while the late region units (L1 to L5) encode proteins that form the viral capsid [24,25,26]. 

Three generations of AdVs have been developed through the deletion of various viral genes. First-generation AdVs were produced by deleting the E1 and/or E3 genes, which leaves these vectors unable to replicate [27,28]. In recent years, first-generation AdVs have shown promise in the development of vaccines against a range of infectious diseases, including influenza, Ebola, and COVID-19 [29,30,31]. Second-generation AdVs have E2 and/or E4 genes deleted along with E1 and/or E3. This not only increases the carrying capacity, but also reduces the potential cytotoxic effects that may come with viral gene expression [32,33]. Third-generation AdVs, also known as gutless or helper-dependent AdVs (HDAdVs), have all viral genes deleted from the viral genome. HDAdVs are attractive as vectors for gene therapy because of their higher transgene capacity, prolonged transgene expression, and a diminished immune stimulation [27,34]. 

## 3. Adenoviruses as Vaccine Vectors 

Traditionally, vaccine design has focused on use of either attenuated versions of a particular pathogen or protein subunits. Although these have provided protection against a variety of life-threatening diseases, next-generation vaccine technologies have introduced the use of nucleic acids and viral vectors as good candidates [35]. Research into the use of Ads as vaccine vectors has been appealing to scientists and researchers for over 30 years. While the ability of Ads to induce both an innate and adaptive immune response in the hosts is not ideal for most therapeutic applications, this feature is useful for vaccine design [36,37]. Innate immune responses, produced soon after infection, are not antigen-specific and do not induce immunological memory. However, an AdV-induced adaptive immune response results in activation of B and T cell differentiation with the subsequent generation of immunological memory [38]. Conventional vaccines generally require adjuvants for activation of innate immunity. By contrast, AdVs contain structural components that are recognised by pattern recognition receptors (PRRs) to activate a robust innate immune response [39].

Although there is a paucity of information on the use of AdVs for developing anti-HBV vaccines, AdVs have successfully been developed as vaccines against other viruses. In a study conducted by Gao et al., the effect of an Ad5-based vaccine encoding hemagglutinin (HA) from the avian influenza virus H5N1 strain isolated from the 2003–2005 outbreak in Vietnam was evaluated (Table 1). Vaccination with full-length HA-encoding Ads induced cellular and humoral HA-specific immunity in mice and conferred protection against viral challenge. Additionally, considering the role that poultry plays in transmission of H5N1, the efficacy of this Ad5-based immunisation was tested on domestic chickens. It was found that all subcutaneously immunised birds that were boosted upon viral challenge were protected against infection [40]. Recently, the use of other AdV serotypes for vaccine design has gained attention, including Ad26, another human serotype, and ChAdOx1, a replication-incompetent simian AdV. Anywaine et al. performed a Phase I randomised clinical trial to evaluate the safety, tolerability, and immunogenicity of an Ad26-derived Ebola vaccine (Ad26.ZEBOV) in a heterologous two-dose regimen in adult volunteers from Tanzania and Uganda (Table 1). The results revealed that 21 days after the second vaccine dose, 100% and 87–100% of the participants demonstrated Ebola virus glycoprotein antibody responses and neutralising antibody responses, respectively [41]. The recent success of AdV vaccines against severe acute respiratory syndrome coronavirus 2 (SARS-CoV-2) has revived interest in using these vectors to protect against other viral infections.

### Development of AdV-Based COVID-19 Vaccines and the Impact on Anti-HBV Vaccine Design

The recent COVID-19 pandemic has exacerbated the global demand for effective and rapid vaccine production. Hundreds of millions of dollars have now been invested in traditional vaccine technologies, such as live-attenuated and protein subunit vaccines, as well as mRNA and viral-vector-based vaccines [42,43,44]. SARS-CoV-2 is an enveloped virus consisting of a positive-sense RNA genome encoding Envelope, Membrane, Nucleocapsid, and Spike (S) proteins. Importantly, S binds to angiotensin-converting enzyme 2 (ACE2) and mediates viral entry into cells [44,45]. Replication-incompetent AdVs encoding the SARS-CoV-2 S protein have been essential to the SARS-CoV-2 pandemic response [37]. Easy adaptability, manufacturing, and storage capabilities make AdV vaccines suited to rapid response in a pandemic situation [35]. 

In 2020, at the height of the pandemic, a study published by Feng et al. revealed that a replication-defective recombinant Ad (Ad5-S-nb2) expressing S, induced S-specific antibody production and cell-mediated immune responses in rodents and nonhuman primates (Table 1) [46]. As the pandemic progressed, AdV-based vaccine research remained at the front line. The chimpanzee adenoviral vector (ChAdOx1)-based ChAdOx1 nCoV-19 vaccine (AZD1222), encoding the SARS-CoV-2 S protein, induced both humoral and cell-mediated immune responses, protection against lower respiratory tract infection in nonhuman primates, an increased spike-specific antibody response by day 28, and a neutralising antibody response after a booster dose in human participants (Table 1) [47]. An overall vaccine efficacy of 70.4% was observed in participants of a randomised, controlled trial [48]. Administration of the commonly known Janssen Ad26.CoV2.S vaccine, encoding the full-length SARS-CoV-2 S protein, showed that a single dose of Ad26.CoV2.S produced S-binding neutralising antibodies and strong humoral immune responses in the majority of the 805 vaccinated participants. It induced a 66.9% and 76.3% protective efficacy across participants of all age groups and in participants over the age of 60, respectively (Table 1) [17,49]. To avoid pre-existing immunity and allow vector re-administration, Lugonov et al. performed a prime boost phase 3 clinical trial using the Ad26- followed by the Ad5-derived vector expressing the full-length S protein. This regimen was well tolerated and 91.6% vaccine efficacy was observed at 21 days after the first dose with the Ad26 vector [19]. As clinical research into adenoviral vaccines for SARS-CoV-2 continues, it is likely that this vaccination technique will be adopted for immunisation against an array of pathogens in the future. One such infectious disease is hepatitis B.

**Table 1 genes-13-01941-t001:** Commonly used AdVs in anti-viral vaccine and immunotherapeutic development.

Ad Vector	Target	Antigen Delivered	Key Findings	References
Ad5	Avian Influenza (H5N1)	Hemagglutinin (HA)	Cell-mediated and humoral HA-specific immunity in mice Protection against viral challenge in mice and chickens	[40]
	SARS-CoV-2	Spike protein	Spike-specific cell-mediated and humoral responses in rodents and nonhuman primates	[46]
	HBV	Fusion protein including modified HBV core, polymerase, and envelope proteins	Production of HBV-specific splenic and intrahepatic T-cells Cytokine production and induction of cytolysis Reduction in circulating virus	[50]
Ad26	Ebola	Ebola virus glycoprotein	Glycoprotein-specific and neutralising antibody responses	[41]
	SARS-CoV-2	Spike protein	Production of Spike-binding neutralising antibodies Strong humoral immune response	[49]
ChAdOx1	SARS-CoV-2	Spike protein	Humoral and cell-mediated immune responses Protection against lower respiratory tract infection in nonhuman primates Increased spike-specific antibody responses Neutralising antibody responses	[47]
	HBV	Three full-length HBV antigens, including precore/core, polymerase, and surface	Enhanced T-cell responses in immunocompetent uninfected mice	[51]

While AdV-based vaccine research has made meaningful contributions to immunisation against SARS-CoV-2, it is important to note that there is a significant gap in application of AdVs to vaccination against HBV. There has, however, been progress in the fields of immunotherapeutics and therapeutic vaccine strategies using AdVs against HBV. This includes stimulating the immune system, after viral infection, to produce an antiviral response. For example, TG1050 is an Ad5-derived novel anti-HBV immunotherapeutic encoding a fusion protein comprising modified HBV Core, Polymerase, and select domains of Envelope proteins [50] (Table 1). A single dose of TG1050 induced splenic and intrahepatic HBV-specific T cells that produced cytokines and stimulated cytolysis, with the resultant reduction in circulating viral replication markers [50]. Recently, Chinnakannan et al. published their findings on the design and development of therapeutic ChAdOx1 and modified vaccinia Ankara (MVA) viral vectors against HBV (Table 1). Administration of ChAdOx1 encoding an HBV immunogen consisting of three full-length HBV antigens (precore/core, polymerase, and surface) followed by a heterologous MVA-boost vaccine produced enhanced T cell responses in immunocompetent uninfected mice. Polyfunctional CD8+ and CD4+ T cells produced cytokine combinations, including IFNγ, TNF-α, and IL-2 [51]. The success of anti-HBV AdV-based immunotherapeutics further emphasises the potential of AdVs to deliver HBV antigens to cells successfully. Additionally, as it has been established that AdV vaccines can induce both cell-mediated and humoral immune responses and that the basic method for production of these vaccines is easily adaptable, research into the use of AdVs to immunise against HBV is currently underway and is important for the advancement of HBV vaccine research. 

## 4. Adenoviruses as Vectors of Gene Therapy against HBV

Because of their high liver tropism, AdVs are attractive for the development of gene therapies targeted to HBV. The reduction in HBV replication markers by higher than 90% in culture and in mice using first-generation AdVs expressing RNAi activators has been demonstrated [52]. However, strong vector-induced immune responses and short-term therapeutic effects in vivo have been observed. Delivery of anti-HBV RNAi activators using HDAdVs resulted in a prolonged therapeutic effect in vivo [53,54]. The dependence of viral persistence on functional cccDNA makes this HBV replication intermediate an ideal target for achieving a sterilising cure from HBV infection. cccDNA targeting using TALENs or CRISPR/Cas9-based therapies has shown promise [55,56,57,58]. These gene editors mediate sequence-specific deleterious mutations within the HBV DNA [22]. Although AdV-based anti-HBV TALEN delivery is not encouraging [59], promising outcomes when using AdVs to deliver anti-HBV CRISPR/Cas sequences have been reported [60,61].

A recent study by Kato et al. successfully demonstrated the efficacy of an AdV expressing a CRISPR/Cas9 system with eight guide RNAs (gRNAs) targeting the HBV X gene (*HBx*) [60]. In this study, use of an AdV was ideal as the increased carrying capacity of the AdV allowed for inclusion of 8 gRNAs and a large Cas protein-encoding sequence isolated from *Streptococcus pyogenes*, spCas9. This Cas9 protein recognises the protospacer adjacent motif (PAM) sequence 5`-NGG-3`, which is more common within the HBV genome. spCas9 is, therefore, preferred to *Staphylococcus aureus*-derived Cas9 (saCas9), which, although smaller than spCas9, uses the PAM sequence 5′-NNGRRT-3′, which is uncommonly found in the HBV genome [61]. A study by Schiwon et al. compared the efficacy of TALENs targeting HBV to a CRISPR/Cas9 system with three different gRNAs, delivered by an HDAdV [59]. The anti-HBV activity of the two gene editors was tested in both HepG2.2.15 and HepG2-NTCP cells. HepG2.2.15 cells contain stably integrated replication-competent HBV sequences [62], and HepG2-NTCP cells express human NTCP, which makes them susceptible to HBV infection in culture [63]. The levels of viral protein, RNA, and genome equivalents were compared. The results showed that the CRISPR/Cas9 system induced a far more significant reduction in viral protein secretion than the TALEN system, with a reduction of 54% and 45% in HBsAg and HBeAg secretion compared to no reduction with the TALEN system, respectively. This was supported by the 64% reduction in HBV transcripts seen in the CRISPR/Cas9-treated cells [59]. This AdV-mediated multiplex approach has therapeutic potential for targeting a diverse range of HBV genotypes or different targets to avoid viral escape, using one vector. Although the efficacy of AdV-based gene therapy is undeniably promising and HDAdVs reduce vector immune stimulation, capsid-induced innate immune response and pre-existing capsid-specific immunity remains a challenge. This has limited success on AdVs in gene therapy clinical trials [64,65]. The recent widespread use of AdV-based COVID-19 vaccines will further exacerbate the challenge of pre-existing immunity.

### Overcoming AdV Immunity in AdV-Based Anti-HBV Gene Therapy

Extensive efforts have been put into developing strategies to overcome immune stimulation and avoid AdV clearance (Figure 1). Immune suppression before vector administration has been explored with promising outcomes [66,67]. However, this method is not favoured, because of the possibility of increased risk of infection by other pathogens. The possibility of using polyethylene glycol (PEG), a synthetic biocompatible compound, to diminish capsid-induced innate and adaptive immune responses has been heavily researched for targeting hepatitis B and other diseases. ‘PEGylated’ vectors have been shown to evade pre-existing immunity, mediate prolonged transgene expression, and reduce AdV toxicity in vivo [68,69,70]. Although multiple studies have shown that PEGylation can shield capsid proteins essential for viral entry, therefore reducing transduction efficiency [70,71,72], several other studies have illustrated that with optimised PEGylation approaches, transduction efficiency remains optimal in vivo [73,74,75]. The genetic modification of capsid epitopes by mutagenesis or direct clonal evolution is another strategy that may be used to generate clonal mutants that evade host immunity [76,77,78]. Importantly, some genetic modifications of the capsid can affect vector structural integrity and diminish transduction efficiency [79,80], while others have minimal effects on vector viability or infectivity [81]. These findings illustrate that understanding the biology of AdV and vector–host interactions are key to the development of chemical or genetic capsid modifications that yield highly efficient and safe AdVs. Using adenoviral serotypes with low seroprevalence in humans, such as Ad26, Ad11, and Ad35, present another alternative to avoiding pre-existing immunity. Several studies have shown that transduction efficiencies of these low-seroprevalence vectors are not affected by the immunity against Ad5, a highly prevalent serotype in human infections [82,83,84]. Although most of these strategies are yet to be tested for hepatitis B gene therapy, they can be easily translatable for the design of highly potent and safer anti-HBV therapeutics.

## 5. Conclusions

AdV’s higher transduction efficiency, which translates to a low dose requirement, easy manufacturing, and low storage costs, amongst other advantages, makes them a feasible option to combat viral infections, especially in low-income countries. Although strong immune stimulation by AdVs can prime a robust and long-lasting adaptive response, which is attractive for vaccine development, this feature may result in rare but severe adverse effects [85,86,87]. Significant progress has been made with diminishing AdV-mediated immune induction. Approaches that include chemical and/or genetic modifications of the capsid and deletion of all viral genes are not enough to avoid the AdV immunity obstacle. Trade-offs between immune evasion and loss of biological functionality also pose a challenge. Approaches that combine multiple strategies, e.g., chemical and genetic manipulation of the capsid, may offer a better solution [78,88,89]. For the foreseeable future, using low-seroprevalence Ads such as Ad26 and ChAdOx1 may be the best strategy to overcome pre-existing immunity.

## Figures and Tables

**Figure 1 genes-13-01941-f001:**
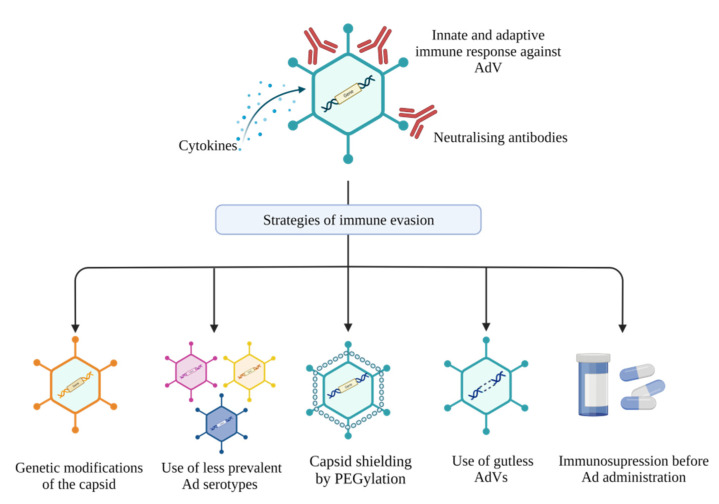
Strategies to avoid immune responses to adenoviral vectors (AdVs). AdVs may elicit both innate and adaptive immune responses to vector proteins. Strategies to evade these immune responses include genetic modification to the viral capsid, use of less prevalent Ad serotypes, shielding the viral capsid by PEGylation, removal of viral sequences to generate gutless AdVs, and administering immunosuppressive drugs before vector injection (created with Biorender.com).

## Data Availability

Not applicable.

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
