# Peer review of "Adenoviral Vectors: Potential as Anti-HBV Vaccines and Therapeutics"

_genes, 2022, doi:10.3390/genes13111941_

Round 1

Reviewer 1 Report

In general, this is a well-prepared paper. Only have one major comment and one minor suggestion. 

1. There is a section of “Development of AdV-based COVID-19 vaccines and the impact on anti-HBV vaccine design”, which included the vaccine development. AdV-Covid 19 vaccines have been world-widely used in prevention of Covid pandemic. It would be better to include the results of the use of Ad-based vaccine in human, compare the results with that of other covid vaccines (mRNA, viral protein, et al), and discuss the advantages and disadvantages of AdV-based Covid vaccines based on the clinical results and comparison. How may the specific clinical findings affect the development AdV HBV vaccines?

Minor comments:

2. There are 3 generations of Ad vectors and the authors provided references for the first and third generations of the vectors, missing the references for the second-generation vectors.

Reviewer 2 Report

Chronic viral hepatitis B (HBV) remains a major global health problem. Currently, about 2 billion people are infected with HBV, and more than 350 million people suffer from chronic hepatitis B (CHB) worldwide. Cirrhosis and hepatocellular carcinoma (HCC) develop in 15-40% of CHB patients without appropriate treatment, and about 1 million patients die each year from cirrhosis, liver failure, or HCC due to chronic HBV infection.

Approved HBV treatment regimens are limited to interferon and nucleoside analogs, but these drugs can only effectively suppress viral replication without killing the virus.

The first hepatitis B vaccine, marketed in 1982, consisted of HBsAg isolated from the plasma of people with chronic HBV infection. This vaccine has now been replaced by recombinant vaccines that do not cause any problems associated with human blood products. Despite the availability of a vaccine to prevent hepatitis B virus transmission, complications resulting from chronic infection with the virus and reduced vaccine efficacy in people over 40 years of age and in those who are immunocompromised remains an important global public health problem.

Adenovirus vectors (AdV) are one of the most popular technological platforms for the development of preventive and therapeutic drugs. The first AdVs for gene delivery were created long ago, in the late 1980s and early 1990s. Over the past 20 years, hundreds, if not thousands, of articles describing experimental preparations based on non-replicating adenovirus vectors have been published.

The advantage of the system based on recombinant adenoviruses is the well-studied safety of this technological platform. To date, more than 500 clinical trials of AdV have been conducted worldwide for use in gene therapy and as vaccine preparations. The high efficacy of approved AdVs vaccines against Ebola and COVID-19 makes the development of a similar vaccine against HBV urgent.

The peer-reviewed paper details the advantages of adenoviral vectors, as well as the prospects for their use in the development of drugs for the prevention and gene therapy of hepatitis B.

The authors analyze all possible complications that may arise during the development of such drugs, in particular, in the event of an immune response to the vector or in the presence of preexisting antibodies to adenovirus. In this regard, another way to avoid the appearance of unwanted antibodies to the vector, based on prime-boost immunization with two different vectors (Ad26 and Ad5) having the same target gene, could be noted by authors. Such a scheme was successfully implemented in the Sputnik V vaccine (Logunov DY et al, Safety and immunogenicity of an rAd26 and rAd5 vector-based heterologous prime-boost COVID-19 vaccine in two formulations: two open, non-randomised phase 1/2 studies from Russia // Lancet 2020 Sep 26;396(10255):887-897 doi: 10.1016/S0140-6736(20)31866-3). A similar approach could be applied not only to vaccines but also to anti-HBV gene therapy.

Summing up, the review is of interest to a wide audience and can be published.
